# OpenReview forum: "DISCO: Disentangled Communication Steering for Large Language Models"
_NeurIPS.cc/2025/Conference — NeurIPS 2025 poster_

### Official Review · Reviewer_X2zd · 2025-06-25

**Clarity:** 4
**Significance:** 3
**Originality:** 4
**Rating:** 5
**Confidence:** 4

**Summary:**

The paper introduced steering vectors for the query and value representation spaces. Furthermore, the method disentangles the steering hyperparameter $\alpha$ used to steer Query and Value vectors, demonstrating better performance using this approach on the TruthfulQA, and other datasets to evaluate communication steering.

**Questions:**

Questions:

1. What does the linear representation space for the attention head output of the keys look like (Figure 1 a(ii) but for keys)?
2. Does the cardinality of $D^+$ and $D^-$ need to be the same? Or are more negative examples required?

**Ethical Concerns:**

["NO or VERY MINOR ethics concerns only"]

**Final Justification:**

I have reviewed the rebuttal and I recommend the paper for acceptance.

**Limitations:**

Yes

**Paper Formatting Concerns:**

1. L143: it’s $\rightarrow$ its
2. The authors could consider replacing the cardinality of the vocabulary $|V|$ with $|\mathcal{V}|$ to avoid confusion with the Value matrix $V$
3. L197: effected $\rightarrow$ affected

**Quality:**

4

**Strengths And Weaknesses:**

Strengths:

1. The main claims of the paper are well supported
2. The authors verify claims across multiple datasets with Llama and Gemma LLMs
3. The mathematical claims are sufficiently proved

Weaknesses:

1. Figure 1 could be simplified and presented early on in the paper with an example to demonstrate the approach better, the figures are difficult to parse on the first read
2. The paper assumes the MHA scenario where queries, keys, and values all have the same number of heads. Some discussion of how this might apply to grouped query attention used in Llama 3 and Gemma 2
3. Some minor typos that could be fixed, see paper formatting concerns

---

> ### Author Rebuttal · Authors · 2025-07-30
>
> We appreciate your positive evaluation of our work, including the breadth of experiments and the correctness of our mathematical results. We address your comments below.
>
> **W1: Figure**
>
> ---
>
> Thank you for your recommendation regarding Figure 1, which we plan to take into account for the 10-page camera ready version. While we do not have the option to include an updated pdf here, we will:
>
> * Recontextualize a(i) and b(i) as parts of a larger network. We will move this portion of the figure below a(ii), b(ii) and (c). This will allow us to declutter and enlarge each element of the figure.
>
> * Using this formatting, include a question + partial generation below the network, as examples of inputs. We will also show a predicted token (chosen to be related to the concept) coming out of the network to illustrate the effect of steering.
>
> * Reference this new example early on in the introduction, and, formatting permitted, move the figure itself up to the top of Page 2.
>
> We also plan to enlarge the other figures and increase font sizes where needed.
>
> **W2: Grouped Query Attention**
>
> ---
>
> We agree that this is a useful distinction to clarify. Conceptually, the application of DISCO in the Grouped Query Attention (GQA) scenario is almost the same as for Multi Head Attention (MHA). The main difference is that the steering vector for a given value space will affect all of the attention head outputs (in the layer) which are a function of that value space. We will include a discussion of this in the main body, and add an additional algorithm in the Appendix illustrating DISCO usage for GQA.
>
> **W3: Minor typos**
>
> ---
>
> Thank you for your careful reading of the paper. We will correct these typos in the camera ready version.
>
> **Q1: Linearity of key representations**
>
> ---
>
> We agree that this is a natural question arising from Figure 1 a(ii) and b(ii). While we do not steer keys due to the invariance shown in Prop. 1, we find that **they exhibit similar linear discriminability to queries and values**. Specifically, Figures 4 and 5 in Appendix E (included in the supplementary submission, as companion figures to Figure 2) show that a significantly higher portion of key spaces are linearly discriminative (with respect to concepts) compared to attention head output spaces, for both LLaMA and Gemma. Following your comments, **we will include a similar companion figure for a(ii) and b(ii), but with keys, in the camera ready Appendix**. We believe that this is a potentially useful finding for interpretability researchers in future works.
>
> **Q2: Cardinality for vector estimation**
>
> ---
>
> The cardinality of $D^+$ and $D^-$ may be different or the same. One type of dataset for which $|D^+| = |D^-|$ are those which consist of questions which are each paired with one positive and one negative answer. This is the case for the corrigibility, wealth-seeking and power-seeking datasets [1] used in our and prior works, such as Contrastive Activation Addition (CAA) [2]. Other work partitions unpaired data from a pool to have $|D^+| = |D^-|$ [3]. However TruthfulQA [4], which we also use, consists of questions with a variable number of positive and negative answers, and happens to have $|D^-| > |D^+|$. Our intuition is that mean-difference vectors are reasonably robust to imbalance in $|D^-|$ and $|D^+|$, but an exploration of the effect of such imbalances would be a valuable direction for future work, complementary to ours and others in the sub-field.
>
> # References
>
> [1] Perez, Ethan, et al. "Discovering language model behaviors with model-written evaluations." ACL, 2023.
>
> [2] Rimsky, Nina, et al. "Steering LLaMA 2 via contrastive activation addition." ACL, 2024.
>
> [3] Arditi, Andy, et al. "Refusal in language models is mediated by a single direction." NeurIPS, 2024.
>
> [4] Lin, Stephanie, et al. “TruthfulQA: measuring how models mimic human falsehoods”. ACL, 2022.

---

> > ### Comment · Reviewer_X2zd · 2025-08-07
> >
> > Thanks for your comments, I retain my score. I believe it is an accurate evaluation of the work.

---

### Official Review · Reviewer_FSMR · 2025-07-02

**Clarity:** 4
**Significance:** 4
**Originality:** 3
**Rating:** 5
**Confidence:** 2

**Summary:**

Steering vectors are a linear direction that are most related to a concept (increases in the direction of the steering vector). This paper aims to expand the toolbox of steering vectors: proposing, characterizing and validating the steering of the query and value spaces internal to attention heads. Paper shows concepts are linearly discriminable in these spaces, and a larger proportion of them have linear discriminability compared to traditional attention head output spaces. Show that the approach disentangles a strong baseline – referred to as Communication steering.

Contributions:
1)	Propose DISCO steering and variants DISCO-Q, DISCO-V and DISCO-QV (query and value).
2)	Analytically characterize the effects and show that it disentangles a strong baseline – steering attention head inputs – enabling finer control.
3)	Show q and v spaces exhibit linear concept discriminability with higher proportion than attention head outputs, and that DISCO Steering is best on LLaMA 3.1 8B and Gemma 2 9B.

**Questions:**

See above.

**Ethical Concerns:**

["NO or VERY MINOR ethics concerns only"]

**Final Justification:**

Thanks for the rebuttal, it addresses my concerns.

**Limitations:**

Yes.

**Paper Formatting Concerns:**

Presentation issue: many figures contain unnecessarily small font - particularly Fig 3, but also others.

**Quality:**

4

**Strengths And Weaknesses:**

Strengths:

1)	The proposal is the first detailed investigation of steering vectors in the Q and V focusing on the inputs to attention rather than the outputs.

2)	Proposes Q, V and QV steering.

3)	Extensive results compare against the existing ways of performing steering as well as multiple other places where steering may be performed show stronger performance over a range of tests on LLaMa 3.1 8B and Gemma 2 9B.

4)	Paper is well-written and clear.

Weaknessess:

1)	Although the proposal has clear novelty, the idea seems to boil down to investigating steering approaches in different places in the transformer. (Though there is a proof of a disentanglement proposition.)

---

> ### Author Rebuttal · Authors · 2025-07-30
>
> Thank you for your positive evaluation of our methods, experiments and the clarity of our work.
>
> **Weaknesses**
>
> ---
>
> As you note, our experiments compare steering queries and values to a number of representation spaces in the model. Aside from proposing to steer queries and values, demonstrating the empirical success of such an approach, and the disentanglement proposition (Prop. 2) which you mention, we wish to briefly highlight two additional novelties in our work.
>
> * **(Unique interpretation of query steering)** In addition to showing how value steering affects attention head output, Prop. 1 illustrates an invariance to key steering and, importantly, a unique interpretation of query steering. Specifically, **steering the query re-weights attention towards tokens in-context that have keys that align with the (query) steering vector**. While steering other representation spaces can indirectly affect the attention weighting in later layers, steering the query is a direct way to draw relevant information from the context.
>
> * **(High portion of discriminative spaces)** We believe that our robust experimental finding that a significantly higher portion of query and value spaces are linearly discriminative (with respect to a concept) than attention head outputs is valuable (as shown in Figure 2). This is a motivating piece of evidence for DISCO, as the linear separability of concepts is strong motivation for the use of steering vectors themselves [1,2]. We additionally believe that this finding may be of general interest to the community, and hope that it inspires future work in steering and potentially interpretability.
>
> **Formatting**
>
> ---
>
> Thank you for pointing out the small font in some of the figures, particularly Figure 3. In the camera ready version, we will increase the font size across all figures where needed and enlarge the figures themselves to improve readability.
>
> # References
>
> [1] Park, Kiho, et al. “The Linear Representation Hypothesis and the Geometry of Large Language Models”. ICML, 2024.
>
> [2] Li, Kenneth, et al. “Inference-time intervention: Eliciting truthful answers from a language model”. NeurIPS, 2023.

---

> > ### Comment · Reviewer_FSMR · 2025-08-05
> >
> > I remain supportive of this paper. It is interesting work.

---

### Official Review · Reviewer_BVpR · 2025-07-02

**Clarity:** 3
**Significance:** 2
**Originality:** 3
**Rating:** 4
**Confidence:** 4

**Summary:**

This paper focuses on the Representation Engineering (RepE) in large language models and proposes a new method named DISCO. Instead of steering the output of attention heads, DISCO steers the representation spaces of queries and values within the attention heads. Experimental results show that DISCO makes concepts more linearly discriminable and demonstrates it to be a more effective approach.

**Questions:**

See the weaknesses mentioned above.

**Ethical Concerns:**

["NO or VERY MINOR ethics concerns only"]

**Final Justification:**

The authors have provided detailed explanations addressing my concerns, and I believe they have clarified the issues to a certain extent. However, I maintain that the current positive rating is already relatively high and not suitable for further elevation. This judgment is based on my comprehensive evaluation of the work's overall value and significance.

**Limitations:**

Yes.

**Paper Formatting Concerns:**

No format problem.

**Quality:**

3

**Strengths And Weaknesses:**

Strengths:

1. The authors propose a method of steering within the query and value representation spaces inside attention heads.
2. The proposed method is easy to understand and straightforward to implement.
3. The paper achieves better performance than baseline methods and thus offers valuable insights for future research.

Weaknesses:

1. As mentioned in the paper, steering methods rely on the linear representation hypothesis, which means certain problems may not be amenable to such approaches. DISCO inherits both the advantages and the limitations of this family of methods.
2. While steering the query and value representations proves effective, it may not be particularly surprising. Since there are many potential locations in the model where steering could be applied, one could systematically explore these to discover better-performing positions. It is even possible that combinations of different locations could outperform steering just queries and values, suggesting that automated discovery might be more effective than manual exploration.
3. The paper would benefit from a more detailed analysis of cases where DISCO does not achieve optimal performance, helping to clarify its boundaries of applicability and inherent limitations.

---

> ### Author Rebuttal · Authors · 2025-07-30
>
> We appreciate your positive reception of our work, its clarity, and our experimental results. We respond to your comments below.
>
> **W1: Linear representations**
>
> ---
>
> This is a reasonable point regarding the limits of steering vector approaches in general. While we agree that certain tasks may be less amenable to this family of methods, we wish to highlight that such methods have been found effective for a very wide variety of concepts/tasks including but not limited to: refusal [1], truthfulness [2], instruction following [3], alignment-related personas [4] and, more recently, reasoning control [5,6].
>
> **W2: Steering locations**
>
> ---
>
> Thank you for your thoughtful comments, which we address below through two connected points.
>
> **(DISCO-V is an especially strong location)** We wish to highlight, as shown in Table 1, that steering DISCO-V on its own outperforms all 7 non-DISCO baselines in 8/16 experiments, and outperforms Inference Time Intervention (ITI) [2], which selects from the attention head output spaces to steer, in 13/16 experiments. We believe this illustrates the utility of the value space, as a component.
>
> **(Expanding on automated discovery)** While outside the scope of our work, your idea to explore automated discovery is interesting. We believe that, while DISCO itself is valuable as an effective steering method, **the empirical and theoretical findings in our work may inform and inspire future research**. From this perspective, we wish to highlight **three ways in which we believe DISCO could help inform this specific future approach you propose**:
>
> * **Value inclusion due to performance:** The strong performance of our proposed value steering (as explained above) stresses the importance of including the value representation spaces in such a search.
>
> * **Efficiency improvement:** Prop. 1 illustrates invariance toward steering the keys while Prop 2. illustrates that steering queries and values disentangles (and thus subsumes) steering the representation that is input to the attention operator. Thus, **these representation spaces can be safely excluded from the search, potentially saving a significant amount of time and computation**.
>
> * **Query inclusion due to uniqueness and interpretability:** We show in Prop 1. that our proposed query steering has a unique interpretation as re-weighting attention toward tokens which have keys that align with the (query) steering vector. Thus, following our work, including the query representation space in the search is important due to its unique functionality. Additionally, this perspective may add some interpretability to the search results. For instance, selection (or non-selection) of query spaces could indicate the utility of such a re-weighting, which may have implications regarding the model and dataset (e.g., if the prompts in the data contain tokens which can be attenuated to promote the concept of interest in the model). While further investigation is needed to make this insight actionable, we believe that it holds meaningful potential.
>
> **W3: DISCO boundaries**
>
> ---
>
> **(Steering vector boundaries)** Thank you for your valuable suggestion. As you note, certain problems/concepts which do not have linearly discriminable features may not be an optimal fit for steering vector approaches in general, including DISCO. We will highlight this in the limitations and future work section of the camera ready version. That said, we note, as discussed in the W1 response, that steering vectors have proven useful for a wide variety of concepts.
>
> **(DISCO boundaries)** A DISCO method achieves the best performance in 13/16 experiments and at least second best in 15/16. This consistency makes it somewhat difficult to pinpoint failure cases that could meaningfully characterize the general limitations of the DISCO framework as a whole. In light of this, we believe that the most informative direction would be to **analyze the potential limitations of DISCO-Q in particular**. This is due to its **unique interpretation** as re-weighting the attention to tokens in context (shown in Prop. 1 and discussed above). This suggests that **DISCO-Q may be particularly effective for questions which contain tokens that have information about the concept, and potentially less effective when such information is absent** (necessitating additional steering in the value spaces).
>
> A rigorous investigation of the hypothesis above is non-trivial, as the notion of a token’s representation “containing information about a concept” is not just a function of the token’s identity, but also of how it is represented by the model of interest at particular layers and heads. Due to the complexity of this interaction and limited time during the review process, we leave a more comprehensive study to future work. That said, **in addition to its theoretical backing from Prop. 1, we do note preliminary experimental evidence for this framing of DISCO-Q’s utility**: DISCO-Q performs particularly well (beating all non-DISCO baselines and DISCO-V, for both models) on the multiple-choice variant of TruthfulQA [7], where answer candidates (which ostensibly consist of very important tokens in terms of answering the question) are included in the context as options. We will include a discussion of the potential limitation of DISCO-Q on questions which do not contain concept-relevant tokens (with respect to a given model) in the limitations and future works section of the camera ready version.
>
> # References
>
> [1] Arditi, Andy, et al. "Refusal in language models is mediated by a single direction." NeurIPS, 2024.
>
> [2] Li, Kenneth, et al. “Inference-time intervention: Eliciting truthful answers from a language model”. NeurIPS, 2023.
>
> [3] Stolfo, Alessandro, et al. “Improving Instruction-Following in Language Models through Activation Steering”. ICLR, 2025.
>
> [4] Rimsky, Nina, et al. "Steering LLaMA 2 via contrastive activation addition". ACL, 2024.
>
> [5] Constantin, Venhoff, et al. “Understanding Reasoning in Thinking Language Models via Steering Vectors.” Reasoning and Planning for LLMs Workshop, ICLR, 2025
>
> [6] Liu, Sheng, et al. “Fractional Reasoning via Latent Steering Vectors Improves Inference Time Compute”. ArXiv, 2025.
>
> [7]  Lin, Stephanie, et al. “TruthfulQA: measuring how models mimic human falsehoods”. ACL, 2022.

---

> > ### Comment · Reviewer_BVpR · 2025-08-06
> >
> > Thank you for your clarifications, which have helped me better understand this work. The authors have specifically addressed the concerns I raised, and I believe they have provided reasonable clarification on these issues. Therefore, I maintain my current positive rating.

---

### Official Review · Reviewer_f7Fy · 2025-07-02

**Clarity:** 3
**Significance:** 3
**Originality:** 2
**Rating:** 4
**Confidence:** 3

**Summary:**

The authors show that injecting steering vectors inside attention heads (queries and values) seems to outperform injecting them simply at their output.

**Questions:**

(Q1) Small comment, but it seems like in Figure 1 DISCO-Q or DISCO-V do not separate better than the attention head output. So perhaps this is not the best example.

(Q2) In Table 1, it might be useful to add an "average rank" or some kind of summary statistic for each method to help better rank them.

(Q3) Do you have some intuition for why V-steering also works? I enjoyed the Q-steering motivation and was wondering if there is some hypothesis on steering the values.

**Ethical Concerns:**

["NO or VERY MINOR ethics concerns only"]

**Final Justification:**

The addition from the authors helps to better convey the message. I believe the method is sound and enjoy some of the motivations in the paper.

**Limitations:**

Yes

**Quality:**

3

**Strengths And Weaknesses:**

The idea is very simple and likely impactful. I enjoyed Proposition 1 which is also rather simple, but provides good motivation for the method (DISCO-Q especially).

Perhaps a weakness is that as far as I can see no motivation is provided for DISCO-V for why it works better than steering the output of the head. The fact that it works better is of course sufficient to write up in a publication, but it would be interesting to have a deeper understanding as it feels like this could provide further insights into the representation mechanisms used by LLMs.

---

> ### Author Rebuttal · Authors · 2025-07-30
>
> Thank you for your positive assessment of our work, we were particularly pleased that you find it likely to be impactful. We respond to your comments below, combining the response for the weakness and Q3.
>
> **Q1: Separability in Figure 1**
>
>
> ---
>
>
> We appreciate your close reading of the plots in Figure 1. To clarify, **our main claim regarding discriminability is that a significantly larger portion of query and value spaces have high linear discriminability compared with the attention head output spaces**. This is in contrast to the most discriminative query/value space necessarily being more discriminative than the most discriminative attention head output space. For instance, in LLaMA3 on the Corrigibility Dataset **15% of query spaces have linear classifiers with $\geq$95% accuracy while only 1.3% of attention head output spaces have classifiers with such accuracy**. While Figure 2 illustrates this superior portion of discriminative heads in query and value spaces (this specific example can be seen visually in the Upper Left Corner subplot), **our goal with Figure 1 was (in addition to illustrating steering) to convey the novel finding that query and value spaces can be linearly discriminative at all.** Following your comment, we will make this distinction more explicit in the camera-ready version of the paper to improve clarity.
>
> **Q2: Average rank**
>
>
> ---
>
>
> We thank you for your idea for conveying our results in a more succinct manner. We provide the average rank results across all datasets below (1 = best, 10 = worst, for a given dataset), which we will include in the camera ready version as the final column of Table 1. We note DISCO-QV and DISCO-V are respectively the first and second most effective methods. The utility of query steering is exemplified by the improved average rank of DISCO-QV over DISCO-V (1.75 vs 3.19) in the table below and the granular results in Table 1, where DISCO-Q attains the best performance in 2 experiments.
>
>
> |               | DISCO-QV | DISCO-V | DISCO-Q | CAA [1] | ITI [2] | Post Attn. | MLP Input | MLP Output | Comm. Steer. | Attn Output |
> |---------------|----------|---------|---------|---------|---------|------------|-----------|------------|--------------|-------------|
> | Avg. Rank     | **1.75**     | *3.19*    | 6.06   | 5.31    | 6.56    | 5.69       | 9.69      | 7.25       | 3.63         | 5.88        |
>
> **Q3 + W1: Interpretation of steering values**
>
>
> ---
>
>
> We appreciate your thoughtful comments and are glad that you enjoyed the motivation for steering the query. We have some hypotheses on why steering the value is effective, although we have not settled on one. From Prop. 1 we see that, when steering the value space, the *value* steering vector pops out of the attention head operator and is effectively added directly to the head output. As we steer multiple heads following the practice in Inference Time Intervention (ITI) [2], one idea is that value steering is particularly effective compared to ITI because a significantly higher portion of value spaces exhibit strong linear discriminability compared to the attention head output spaces used for vector estimation in ITI, yielding more effective steering vectors. However, this is just one possibility and it also offloads the question to another: “Why are a higher portion of value spaces discriminative compared to head outputs?”. We believe that this question, as well as the initial one posed, are both interesting and non-trivial and we leave their answers to future work.
>
> # References
>
> [1] Rimsky, Nina, et al. "Steering LLaMA 2 via contrastive activation addition." ACL, 2024.
>
> [2] Li, Kenneth, et al. “Inference-time intervention: Eliciting truthful answers from a language model”. NeurIPS, 2023.

---

> > ### Comment · Reviewer_f7Fy · 2025-08-01
> >
> > I would like to thank you for your responses. I think the average rank looks quite convincing and perhaps helps to convey the message more clearly. I am happy to keep my score.

---

### Official Review · Reviewer_vLp8 · 2025-07-03

**Clarity:** 3
**Significance:** 3
**Originality:** 3
**Rating:** 4
**Confidence:** 4

**Summary:**

In this paper, the authors propose DISCO Steering, which applies mean-difference “steering vectors” directly to the query (Q) and value (V) sub-spaces inside each attention head, in order to flexiblily control the behavior of large language models (LLMs) at inference time without full fine-tuning. Specifically, the authors (1) prove a straightforward analytical relation showing how Q/V injections re-weight attention scores, (2) note a simple invariance that makes key (K) injections moot, and (3) run empirical tests on two medium-sized open models (LLaMA-3 8 B and Gemma-2 9 B) over four behaviour datasets (TruthfulQA, Corrigibility, Power-Seeking, Wealth-Seeking). On the selected evaluation benchmarks, the proposed solution can give finer control or better results.

**Questions:**

1. DISCO-QV optimises $\alpha_q$ and $\alpha_v$ independently but finally applies the same scalar to every head chosen. Would head-specific magnitudes further help with the capability?

2. The head selection is an important step in the proposed approach. How sensitive of the proposed approach to the selection of heads? Would a different selection of $k$ in top-$k$ selection or even a random selection fully mitigate the performance gain?

3.  Can DISCO simultaneously steer multiple attributes without cross-talk?

4. What happens for contexts of 4k or 16k tokens where attention sparsity patterns differ?

**Ethical Concerns:**

["NO or VERY MINOR ethics concerns only"]

**Final Justification:**

Thank the authors for the detailed response. This has addressed the majority of my concerns and thus, I have adjusted my rating accordingly.

**Limitations:**

The authors have discussed the limitations.

**Quality:**

2

**Strengths And Weaknesses:**

Strength:
- Novel solution to steering: The paper introduces query and value space steering (DISCO-Q/V), moving beyond residual-based methods. This disentangles two roles in attention, selection and content, and generalizes prior methods like Communication Steering. To the best of my knowledge, this is a novel approach for this goal.

- Solid theoretical support: The authors connect their method to the linear representation hypothesis and validate it with linear probe results. Analytical derivations explain how Q and V injections influence attention and output, providing decent theoretical foundation.

- Behavior analysis: The paper shows that decoupling Q and V magnitudes allows stronger steering with less degradation. Case studies and ablations support the effectiveness of targeting top heads and tuning Q/V separately.

Weakness:
My major concern is that the evaluation scope may not be sufficiently comprehensive to validate the generalization capability of this paper. Specifically:
>- **Narrow model coverage**: The paper tests only on LLaMA-3.1 8B and Gemma2-9B, both relatively small models. Whether the method generalizes to larger or more diverse models (e.g., LLaMA-3.1 70B, or even vision language models/coding models) remains unknown, which limits its practical relevance.
>- **Incomplete baseline comparison**: The method is only compared to basic steering approaches like ITI and CAA. Stronger methods such as ReFT [1], AcT [2], and even LoRA [3] are omitted. Although some of these baselines may suffer from larger steering computation, it would be beneficial to see how the steering capability stands among a wider range of model adaptation approaches.
>- **Limited task diversity**. Evaluation is restricted to short, single-turn behavioral tasks judged by GPT-4o. There is no assessment on multi-turn dialogue, long-context reasoning, or other real-world control tasks. This narrow focus weakens claims of general applicability.

[1] Wu, Zhengxuan, et al. "Reft: Representation finetuning for language models." Advances in Neural Information Processing Systems 37 (2024): 63908-63962.

[2] Rodriguez, Pau, et al. "Controlling Language and Diffusion Models by Transporting Activations." arXiv preprint arXiv:2410.23054 (2024).

[3] Hu, Edward J., et al. "Lora: Low-rank adaptation of large language models." ICLR 1.2 (2022): 3.

---

> ### Author Rebuttal · Authors · 2025-07-30
>
> Thank you for taking the time to review our work. We were glad that you found our method novel, and that you appreciated both its theoretical foundation and the utility of our magnitude decoupling.
>
> We have carefully considered each of your points and, where possible, have conducted, or plan to conduct, additional case studies in response (detailed below). We will include these in the camera-ready appendix. In some responses, we reference prior work to highlight some frequent practices in recent Representation Engineering (RepE) [1] steering works. This is intended to offer helpful context for the scope of our evaluations, and not as a dismissal of your feedback, which we appreciate.
>
> **W1: Model coverage**
>
> ---
>
> We recognize your point regarding the value of evaluating methods on larger models, but note that **many RepE papers do not evaluate on models with more than 9B parameters [2,3,4,5].** For instance, the largest LLM steered in AcT [3] is LLaMA 3 8B.  Although we do **not have access to computing resources for re-running all experiments with LLaMA 3.1 70B**, we were able to **reproduce the linear discriminability results from Figure 2 on a 4-bit quantized version of the model**. A significantly higher portion of query (Q) and value (V) spaces are linearly discriminative compared to attention head outputs (H) for the P (Power), C (Corr.), W (Wealth) and T (TruthfulQA) datasets with respect to a variety of minimum accuracy cutoffs, as shown in the table below. We will include a figure in the camera ready appendix.
>
> |Acc.|P||||W||||C||||T|||
> |----|----|--|--|-|----|--|--|-|----|--|--|-|----|--|--|
> ||Q|V|H||Q|V|H||Q|V|H||Q|V|H|
> |≥60%|**.92**|**.96**|.87||**.90**|**.93**|.77||**.97**|**1.0**|.84||**.90**|**.94**|.69|
> |≥70%|**.86**|**.92**|.50||**.72**|**.89**|.33||**.95**|**.99**|.54||**.73**|**.87**|.19|
> |≥80%|**.64**|**.87**|.22||**.16**|**.66**|.09||**.82**|**.86**|.21||.01|**.13**|.01|
> |≥90%|**.09**|**.49**|.05||**.02**|**.10**|.01||**.43**|**.72**|.06||.00|.00|.00|
>
> **W2: Baselines**
>
> ---
>
> We appreciate your suggestion and clarify below how our aims inform the comparisons we include.
>
> **(DISCO framing)** One of the primary aims of this work is to characterize and motivate the use of the query and value representations as **building blocks for steering vector methods**. We end our limitations section stating our **hope that future RepE works, inspired by ours, will incorporate query and value spaces into their design.** Here, we cited AcT [3] and ReFT [6] as respective examples of affine and optimization based approaches which could potentially synergize with our findings.
>
> To **complement this framing of DISCO as a framework using queries and values** as building blocks, **we feel that it is important to include DISCO variants of any non steering vector RepE approach that we compare with**, by applying them in the query and value spaces. **We will include a case-study in the camera ready appendix evaluating AcT and a combination of AcT + DISCO**. We leave evaluation of ReFT and ReFT + DISCO to future work because ReFT’s optimization based nature makes developing an effective methodology for combining it with DISCO potentially non-trivial and time-intensive (in addition to the higher computational demands for estimation, which you allude to).
>
> **(LoRA)** We agree that it would be valuable to compare with LoRA [7], which we did not include (in line with some prior works[2,3]) to focus on the utility of the query and value spaces vs others for steering vectors, and the higher estimation cost (which you note). **We will include a case-study in the camera ready appendix comparing LoRA and DISCO** varying the number of “train” prompts used for vector estimation/training. Preliminary DISCO-V and LoRA results on the TruthfulQA generation task show **DISCO-V outperforms LoRA** with a TxI of **83.1** vs 59.3, and **72.0** vs 34.2 in the small data regime (train set only from 5 questions).
>
> **W3: Tasks**
>
> ---
>
> While we agree that evaluating with complex multi-turn or long-context tasks would be valuable, we believe our experimental scope is **in line with prior work**. Although direct comparisons are difficult, [4, 8,9,10], among others, consist of tasks with comparable complexity. For instance, the main experiments in [10] involve multiple choice and generation for truthfulness and multiple choice for bias reduction.
>
> **We also believe that the commonly used benchmarks [11,12] which we have evaluated sufficiently demonstrate the utility of our approach:** we score the ability to promote and suppress corrigibility, wealth-seeking and power-seeking behaviors in generation as well as promote truthfulness in generation and multiple choice settings. We note that users may wish to interact with models for which the behaviors above are modulated, and the specific importance of truthfulness for many LLM applications.
>
> While we propose a general framework for steering, we note that there exist recent works for which the focus is solely on developing steering vector methodologies for *a specific type of complex task* (e.g., multi-turn chatbot interactions [13] and reasoning [14,15]). We would be happy to see future work build on our DISCO framework in a similar way.
>
> **Q1: Head scalars**
>
> ---
>
> We appreciate this thoughtful question and believe that head-specific magnitudes may further enhance DISCO’s effectiveness.  We elected to use one scalar for each chosen head so as not to add a confounding factor (namely, the head-specific magnitude selection method) in comparing the efficacy of steering different representation spaces. We leave the development of head-specific magnitude selection methods which synergize with DISCO for future work.
>
> **Q2: Head selection**
>
> ---
>
> We agree that understanding the sensitivity to the selected heads is important. To provide some insight, we ran an experiment using DISCO-V to increase the power-seeking behavior on LLaMA 3 with the same number of heads (k=160), but chosen at random rather than the top-k most discriminative. While DISCO-V with the top-k heads achieves a score of 2.98, using the random-k heads scores 2.52. The same experiment run for attention head outputs (ITI [8]) yields a decrease from 2.62 with top-k heads to 2.29 with random-k heads.
> This result provides further validation of the intuitive linear representation hypothesis [16] inspired methodology we have adopted from ITI. While we have not conducted a full sweep over all possible values of k due to compute and financial constraints associated with querying the GPT judge, this result provides evidence that informed head selection does indeed contribute to performance. We will include these results in the camera ready appendix.
>
> **Q3: Multiple attributes**
>
> ---
>
> We conduct a proof of concept case study to assess whether DISCO can steer multiple attributes simultaneously. We apply DISCO-V to Gemma 2 to jointly promote power-seeking and corrigibility. In the interest of time, we scale the previously found $\alpha_v$ values for both concepts vectors by $½$, rather than searching for a strong combination (thus providing a lower bound on performance). We evaluate this combined steering approach on the power-seeking and corrigibility test sets. **The power-seeking score increases from 1.62 to 2.53; the corrigibility score from 1.56 to 2.79**. We find this result promising, and leave a thorough investigation of multi-attribute DISCO steering to future work.
>
> **Q4: Long context**
>
> ---
>
> This is an interesting question, and one that applies broadly to RepE methods (although most works that we are aware of do not evaluate in this setting, e.g., [3, 9, 10]). While we have not evaluated DISCO under long-context settings due to time and compute constraints, we believe that they could potentially be an additional compelling use case for query steering, which we show in Prop. 1 can be used to influence attention patterns. However, this remains an intuition that must be further fleshed out with thoughtful experimentation, which we leave to future work.
>
> # References
>
> [1] Zou, Andy, et al. “Representation Engineering: A Top-Down Approach to AI Transparency” ArXiv, 2023.
>
> [2] Stolfo, Alessandro, et al. “Improving Instruction-Following in Language Models through Activation Steering”. ICLR, 2025.
>
> [3] Rodriguez, Pau, et al. “Controlling Language and Diffusion Models by Transporting Activations”. ICLR, 2025.
>
> [4] Wang, Weixuan, et al. “Semantics-Adaptive Activation Intervention for LLMs via Dynamic Steering Vectors”. ICLR, 2025.
>
> [5] Konen, Kai, et al. “Style Vectors for Steering Generative Large Language Models”. EACL, 2024.
>
> [6] Wu, Zhengxuan, et al. “ReFT: Representation Finetuning for Language Models”. NeurIPS, 2024.
>
> [7] Hu, Edward J., et al. “LoRA: Low-Rank Adaptation of Large Language Models”. ICLR, 2022.
>
> [8] Li, Kenneth, et al. “Inference-time intervention: Eliciting truthful answers from a language model”. NeurIPS, 2023.
>
> [9]  Cao, Yuanpu, et al. “Personalized Steering of Large Language Models: Versatile Steering Vectors Through Bi-directional Preference Optimization”. NeurIPS, 2024.
>
> [10] Qiu, Yifu, et al. “Spectral Editing of Activations for Large Language Model Alignment”. NeurIPS, 2024.
>
> [11]  Lin, Stephanie, et al. “TruthfulQA: measuring how models mimic human falsehoods”. ACL, 2022.
>
> [12] Perez, Ethan, et al. "Discovering language model behaviors with model-written evaluations." ACL, 2023.
>
> [13] Bo, Jessica Y., et al. “Steerable Chatbots: Personalizing LLMs with Preference-Based Activation Steering”. ArXiv, 2025.
>
> [14] Constantin, Venhoff, et al. “Understanding Reasoning in Thinking Language Models via Steering Vectors.” Reasoning and Planning for LLMs Workshop, ICLR, 2025.
>
> [15]  Liu, Sheng, et al. “Fractional Reasoning via Latent Steering Vectors Improves Inference Time Compute”. ArXiv, 2025.
>
> [16]  Park, Kiho, et al. “The Linear Representation Hypothesis and the Geometry of Large Language Models”. ICML, 2024.

---

> > ### Author Response · Authors · 2025-08-06
> >
> > Dear Reviewer,
> >
> > We thank you again for your thoughtful review. As the discussion period draws to a close, we wanted to check whether our responses and additional results have been helpful in addressing your concerns.

---

> > ### Comment · Reviewer_vLp8 · 2025-08-07
> >
> > Thank the authors for the detailed response. This has addressed the majority of my concerns and thus, I have adjusted my rating accordingly.

---

### Decision · Program_Chairs · 2025-09-17

**Decision:**

Accept (poster)

**Comment:**

This paper introduces DISCO Steering, which applies steering vectors directly in the query and value spaces of attention heads, rather than in residual stream. The authors provide results clarifying the effect of such interventions, and validate the approach empirically on LLaMA-3.1 and Gemma-2 across several behavioral steering benchmarks.

The reviewers found the method novel, easy to implement, and supported by clear motivation. They also appreciated the demonstration that query and value spaces contain a larger proportion of discriminative directions than traditional attention head outputs. Weaknesses raised by multiple reviewers were centered on the scope of evaluation.

In the rebuttal, the authors provided additional clarifications, new case studies, and preliminary results, including tests on larger models in quantized form, promissed comparisons with LoRA, and evidence that informed head selection matters. They also clarified limitations. Reviewers acknowledged these clarifications, with several explicitly noting that most of their concerns had been addressed, and all maintained positive scores.

Taken together, the work offers a well-motivated and clearly presented contribution to the growing literature on representation engineering and model steering. The combination of conceptual clarity, novel methodology, and promising empirical evidence makes the case for acceptance.